# Correlation between Cord Blood Vitamin D Levels and Problem-Solving Neurodevelopment in Early Childhood: A Cohort Study in Rural Indonesia

**DOI:** 10.3390/children9101581

**Published:** 2022-10-19

**Authors:** Stephanie Supriadi, Djatnika Setiabudi, Anindita Noviandhari, Raden Tina Dewi Judistiani, Budi Setiabudiawan, Meita Dhamayanti

**Affiliations:** 1Department of Child Health, Hasan Sadikin Hospital, Faculty of Medicine, Universitas Padjadjaran, Bandung 40191, West Java, Indonesia; 2Department of Public Health, Faculty of Medicine, Universitas Padjadjaran, Bandung 40191, West Java, Indonesia

**Keywords:** cord blood vitamin D, infant neurodevelopment, problem-solving domain

## Abstract

Vitamin D influence on brain development and subsequent postnatal neurodevelopment remains controversial. We explored the correlation between cord blood vitamin D levels and longitudinal neurodevelopment in early childhood. A cohort study was conducted on term infants with no congenital abnormalities, born from pregnant women from a cohort study. Cord blood samples were collected to measure vitamin D. Neurodevelopment was examined three times in infants aged 6, 12 and 24 months using the Ages and Stages Questionnaire-3, which comprises 30 questions of five developmental domains: gross motor, fine motor, communication, problem-solving and social–personal. Statistical analysis was conducted with Spearman’s rank correlation and multiple linear regression. Of the 141 babies born from previous cohort studies, only 116 participants were included. The mean level of cord blood vitamin D was 16.2 ng/mL. The percentage participants with vitamin D deficiency and insufficiency were 12.9 and 65.5, respectively. Cord blood vitamin D and the problem-solving domain for infant aged 12 and 24 months were correlated (r = 0.217 and 0.414, respectively). Multiple linear regression showed a decreased problem-solving domain score of 0.641 associated with decreased vitamin D levels. In conclusion, cord blood vitamin D levels correlated with infant neurodevelopmental status.

## 1. Introduction

Vitamin D is a pro-hormone secosteroid with active metabolites of 25-hydroxyvitamin D (25-[OH]D) and 25-dihydroxyvitamin D (1,25-[OH]2D) [1,2]. Vitamin D classically participates in calcium and phosphorus metabolism and thus contributes to bone growth and reorganisation. Nonclassical roles of vitamin D are found significantly at extraskeletal sites and include reducing oxidative stress, antimicrobial defence, immunoregulation and anti-inflammatory, anticancer, neuroprotective and child neurodevelopment roles [1,3,4].

There has been an emerging awareness of hypovitaminosis D as a major health problem worldwide among all age groups [5]. According to the South East Asian Nutrition Survey (SEANUT), levels of serum 25(OH)D < 25 nmol/L (<10 ng/mL) were classified as vitamin D deficiency and levels from 25 to <50 nmol/L (<20 ng/mL) were classified as insufficiency [5].

Several studies have investigated the correlation between maternal vitamin D levels at various gestational ages and newborn vitamin D levels. A previous study noted a positive correlation, although this positive correlation was absent in another study [6,7]. 

The brain starts developing from the beginning of pregnancy and continues into the postnatal period. Various parts of the brain develop at different times and have different critical window periods [8]. A previous cohort study showed that mothers who had vitamin D deficiency during pregnancy reflected lower developmental scores, such as gross motor function being significantly impaired at 3 months of age. As with an association between vitamin D levels in early pregnancy and neurodevelopment outcomes, it seems likely that infant vitamin D status at birth may also be linked with neurodevelopmental outcomes in offspring [7,8]. 

Some observational studies were conducted to analyse the relationship between newborn vitamin D levels taken from cord blood and neurodevelopmental status, with various results. Studies conducted in the United States and Australia demonstrated a positive relationship between cord blood vitamin D levels and early childhood development, including neurocognitive and language aspects [8,9,10]. On the other hand, a study in Shanghai observed no association between cord blood vitamin D levels and all developmental aspects in children aged 2 years [11]. The first 1000 days of life are considered a critical period in which rapid central nervous system development occurs [12]. A disturbance during that period, such as hypovitaminosis D, could create a huge impact on the neurodevelopmental process. Therefore, this study was interested in investigating the correlation between vitamin D levels and the developmental status of children during this critical period. 

## 2. Materials and Methods

### 2.1. Study Design and Sampling 

This cohort study was a part of a larger study on vitamin D status and its impact on pregnancy and childhood in West Java, Indonesia. The study was conducted in the Sukabumi and Waled district, a rural area of West Java Province, Indonesia, from 2016 to 2019. Ethical approval was obtained by the Research Ethics Committee Universitas Padjadjaran. The study participants included infants born from pregnant women noted in the previously mentioned study [7]. The pregnant women were informed about the research by our team at their respective locations. Written consent was obtained from women who had agreed to all research procedures. Cord blood samples were obtained immediately after delivery and then stored and examined for 25(OH)D levels. The developmental status of their child was evaluated at 6, 12 and 24 months. The inclusion criteria were being born at term (37–42 weeks of gestation) and of appropriate size for gestational age. Exclusion criteria were congenital abnormalities, history of bilirubin encephalopathy, seizures and asphyxia. 

### 2.2. Vitamin D Laboratory Measurement

Vitamin D samples were obtained from biological material withdrawn from cord blood and stored between −70 °C and −80 °C. Vitamin D levels were measured using the enzyme-linked fluorescent assay technique with the VIDAS^®^ 25 OH Vitamin D TOTAL measuring device [13]. Vitamin D levels were classified as deficient if serum levels were <10 ng/mL or 25 nmol/L [5]. 

### 2.3. Child Developmental Tools

Child developmental assessments were conducted at the study site at Sukabumi and Waled regional health centres. Mothers recruited into the study were contacted by the research team to schedule follow-ups for the infants at the ages of 6 (V1), 12 (V2) and 24 (V3) months at the regional primary health centres for the assessment of their neurodevelopmental status using the Ages and Stages Questionnaire-3 (ASQ-3). The ASQ-3 was translated into Bahasa Indonesian and has been validated. This tool comprises 30 questions assessing five developmental domains: gross motor, fine motor, communication, problem-solving and social–personal domains. The answers to each question were ‘yes’, ‘sometimes’ and ‘no’ and were scored 10, 5 and 0, respectively. Each domain was then summed. Higher scores reflected better developmental states. These assessments were conducted by researchers and research assistants who had received standardised training [14]. 

### 2.4. Statistical Analysis

Mean, median, range, standard deviations, frequencies and percentages were used to describe numerical or categorical data in descriptive analysis. The Mann–Whitney *U* test or Kruskal–Wallis test was used to determine the differences between the developmental status of children and various aspects based on subject characteristics. The correlation between newborn vitamin D levels and scores for each developmental domain was calculated using Spearman’s rank correlation. Categorical differences between vitamin D levels and each developmental domain were tested using analysis of variance (ANOVA). Multiple linear regression analysis was conducted to adjust all variables tightly correlated with vitamin D levels and external variables. *p*-values of <0.05 were considered statistically significant. Data were managed and analysed using the Statistical Package for Social Sciences (SPSS version 20.0; SPSS, Inc., Chicago, IL, USA) programme. The covariates in this study were maternal age, parity status, maternal educational status and occupation. The characteristics of infants were sex and exclusive breastfeeding.

## 3. Results

Of the 141 babies born from previous cohort studies, 4 were born premature, 10 had intrauterine growth restriction and 11 did not arrive for follow-up until the end of the observation and were excluded. Therefore, there were 116 total participants in this study. The minimum sample size had been fulfilled, with the power of a test of 95%.

Table 1 presents data on the subjects’ characteristics. The majority of mothers were housewives (86.2%) and primipara (72.4%) and had a low educational level (72.43%). The majority of infants were exclusively breastfed (90.5%).

We examined the serum from 116 newborns for vitamin D levels. The mean newborn vitamin D level in this study was 16.27 ng/mL. As many as 12.9% of infants had vitamin D deficiency, 65.5% had insufficiency and 21.6% had normal vitamin D levels (Table 2).

Table 3 shows the mean ASQ-3 developmental scores for each domain at three different times of visit. There was a significant difference in the ASQ-3 developmental scores in the five developmental domains at each different visit (V1, V2 and V3).

A correlation was found between newborn vitamin D levels and the developmental problem-solving domain (Table 4) at the ages of 12 and 24 months (*r* = 0.217 and 0.414, respectively). In other words, the higher the newborn vitamin D levels, the higher the problem-solving scores. This showed that the strength of correlation had increased by 24 months of observation. According to Guilford correlation criteria, this showed a moderate correlation.

Based on the ANOVA test (Table 5), the results showed that vitamin D levels had a significant relationship to the problem-solving domain at the last two visits (V2 and V3). To further prove the relationship between vitamin D levels and the problem-solving domain, a multivariate linear regression analysis was conducted (Table 6). The results showed that a decrease in the problem-solving domain score of 0.641 was associated with a decrease in vitamin D levels.

## 4. Discussion

Hypovitaminosis D in pregnant women and neonates is currently a major concern in obstetrics and neonatology [6]. This study was a part of the initial cohort study on vitamin D status and its impact on pregnancy and childhood in Indonesia [7]. The initial evidence related to the role of vitamin D in brain function was reported two decades ago through the discovery of vitamin D receptors (VDRs) using autoradiography in experimental animal brains and the discovery of 1,25(OH)2D in cerebrospinal fluid [15]. Additionally, VDRs also are widely distributed in mammalian brains and are first expressed in brain development during the critical period of cell proliferation. These receptors can be found in certain brain regions, such as temporal lobes, cingulate, thalamus, cerebellum, amygdala and hippocampus areas [15,16]. 1,25(OH)2D also works by affecting the production of cytokines and affecting neurotransmitters and synaptic plasticity, which have important roles in the learning process and neurocognitive development [16,17]. 

Foetal brain development starts during the early period of pregnancy. Some parts of the brain will develop rapidly in the last trimester, and the process of differentiation and synaptogenesis will develop up to the postnatal period, with the critical time in the first 1000 days of life [17]. Disruptions occurring during this period, such as low maternal vitamin D levels during pregnancy, could lead to impaired foetal brain structure formation, such as brain ventricular enlargement and neocortex region thinning [4,8]. Accordingly, vitamin D deficiency may weaken the integrity of perineuronal nets so that neural circuit function will be disturbed and cognitive processes, such as learning and memory, will be impeded [18]. 

The mean newborn vitamin D level from our 116 participants was 16.2 ng/mL (8.0–35.4 ng/mL). As many as 12.9% of infants had vitamin D deficiency and 65.5% had insufficiency. A previous study on vitamin D levels in the first trimester of pregnancy, which was a part of this cohort study, showed a mean maternal 25(OH)D level of 17.52 ng/mL [7]. This result is similar to those of previous studies showing that newborn vitamin D levels are lower (75–90%) than maternal vitamin D levels [6,9]. This could be occurring because the mother fulfils her own vitamin D requirement before those of her foetus [19]. 

Previous studies conducted to determine the correlation between vitamin D levels during early, middle, late pregnancy and in cord blood at birth and child development were conducted in several countries with various results [4,9,10,20]. The previous portion of this cohort study in Indonesia regarding associations between maternal vitamin D levels in early pregnancy (10–14 weeks of gestation) and child development in the first-year ages of life (3, 6 and 12 months) showed that ASQ-3 scores in gross motor domains were significantly lower at 3 months of age and that there were no significant differences for all developmental aspects at older ages (6 and 12 months) [7,19]. 

We examined cord blood vitamin D samples because maternal vitamin D could pass through the placenta and enter the foetal bloodstream, with a half-life of approximately 2 months. Therefore, maternal levels could represent vitamin D levels in the newborns [21,22]. This study showed a correlation between cord blood vitamin D levels and the developmental problem-solving domain at 12 and 24 months of age, hence no correlations with gross motor, fine motor, communication and personal–social domains. Multiple linear regression analysis reinforces the statement that a decrease in problem-solving scores is associated with a decrease in vitamin D levels. This could be explained by the extraskeletal effect of vitamin D on neuroplasticity, which determined the neurocognitive aspect [16]. The problem-solving domain can represent the neurocognitive aspect. Adverse effects occurring in early life, such as hypovitaminosis D and inadequate stimulations, could cause neurocognitive disturbance that can persist until later life [17]. This was consistent with a recent meta-analysis study regarding the association of maternal or newborn vitamin D levels with neurodevelopmental outcomes, which indicated that prenatal vitamin D levels had borderline positive associations with infant cognitive development but no association with infant motor development [23]. 

A cohort study in the United States assessing the relationship between cord blood vitamin D and developmental and cognitive achievement scores showed that an increase of 5 nmol/L in cord blood vitamin D levels was associated with a minute increase in the Wechsler Intelligence Scale for Children score at the age of 7 years [9,24]. Moreover, a study in Australia stated that cord blood vitamin D levels had a positive association with language development in children aged 18 months and 4 years, although the association was weak [10]. As a result, vitamin D has an effect on child development despite the weak association found in the currently available evidence.

The effort to increase infant vitamin D levels can be made in several ways. The provision of vitamin D supplementation in Indonesia remains difficult as a standard because of the high cost. As an alternative, exposing 18.59% of body surface area to sunlight for 37.5 min/day, especially between 1000 and 1300 h, is proven to be an effective method to meet daily vitamin D requirements [25]. 

The limitations of this study were not reported because several bias factors were difficult to control. For example, parenting style and complementary food given after the age of 6 months could have affected vitamin D levels and neurodevelopmental status at the age of 2 years. Future long-term cohort studies that will consider these factors, or a randomised clinical trial on vitamin D supplementation, are required.

## 5. Conclusions

Vitamin D levels correlate with infant neurodevelopmental status, especially in the problem-solving domain. There was an absence of associations in the other domains. Screening for vitamin D levels during early infancy is crucial for early detection and intervention as well.

## Figures and Tables

**Table 1 children-09-01581-t001:** General characteristics of infants and mothers (*n* = 116).

Characteristics	Total	%
Characteristics of infants		
Sex:		
Male	55	47.4
Female	61	52.6
Exclusive breastfeeding:		
Yes	105	90.5
No	11	9.5
Characteristics of mothers		
Age (years):		
<20	6	5.2
20–29	58	50.0
≥30	52	44.8
Education:		
Low	84	72.4
Average	28	24.1
Well	4	3.4
Occupation:		
Housewife	100	86.2
Employee	16	13.8
Parity:		
Primipara	84	72.4
Multipara	33	27.6

**Table 2 children-09-01581-t002:** Cord blood vitamin D in newborns.

Cord Blood Vitamin D (ng/mL)	*n* (%)
Deficiency (<10 ng/mL)	15 (12.9)
Insufficiency (10 to <20 ng/mL)	76 (65.5)
Normal (≥20 ng/mL)	25 (21.6)
Mean (SD):	16.27 (6.14)
Median (range):	15.25 (8.0–35.4)

**Table 3 children-09-01581-t003:** Mean ASQ-3 scores for each developmental domain at three different times of visit.

Developmental Domains	Time of Visit			*p*-Value *
	6 Months (V1)	12 Months (V2)	24 Months (V3)	
Social–personal	52.03 (7.60)	47.52 (8.69)	44.31 (7.58)	<0.001
Problem-solving	55.04 (6.06)	47.3 (6.89)	49.46 (9.85)	<0.001
Fine motor	54.14 (7.08)	49.57 (6.23)	33.97 (11.13)	<0.001
Gross motor	43.58 (10.10)	48.3 (9.50)	43.43 (11.34)	<0.001
Communication	54.96 (5.17)	52.61 (5.67)	53.82 (10.00)	0.005

* Based on Kruskal–Wallis test.

**Table 4 children-09-01581-t004:** Correlation between newborn vitamin D levels and infant development.

Developmental Domains	Correlation Coefficient (*r*_s_)		
	6 Months	12 Months	24 Months
Social–personal	0.046	0.001	0.180
Problem-solving	−0.083	0.217 *	0.414 *
Fine motor	−0.057	0.105	0.084
Gross motor	0.112	0.183	−0.030
Communication	−0.006	0.001	0.111

*r***_s_**, Spearman’s rank correlation coefficient. * *p* < 0.05.

**Table 5 children-09-01581-t005:** Comparison of infant developmental status at three different times of visit based on vitamin D levels.

Developmental Domains	Cord Blood Vitamin D (Category)			*p*-Value *
	Deficiency (*n* = 15)	Insufficiency (*n* = 76)	Normal (*n* = 26)	
Visit 6 months:				
Social–personal	52.33 (7.76)	51.38 (8.11)	53.80 (5.64)	0.384
Problem-solving	57.00 (5.28)	54.61 (6.36)	55.20 (5.49)	0.375
Fine motor	55.67 (5.88)	53.75 (7.22)	53.80 (7.26)	0.336
Gross motor	46.67 (6.99)	43.62 (10.44)	41.60(10.48)	0.310
Communication	55.33 (4.81)	55.07 (5.13)	54.40 (5.65)	0.820
Visit 12 months:				
Social–personal	46.67 (7.48)	47.93 (9.41)	46.80 (7.20)	0.767
Problem-solving	43.33 (7.94)	47.53 (6.49)	49.00 (6.77)	0.036
Fine motor	49.33 (4.58)	49.40 (6.47)	50.20 (6.53)	0.849
Gross motor	45.00(11.02)	48.20 (9.36)	50.60 (8.70)	0.195
Communication	52.00 (4.14)	53.20 (5.85)	51.20 (5.82)	0.285
Visit 24 months:				
Social–personal	39.64 (11.68)	44.32 (6.67)	47.27 (5.50)	0.012
Problem-solving	43.57 (10.46)	48.56 (9.80)	55.91 (5.70)	<0.001
Fine motor	35.71 (13.13)	32.58 (10.31)	37.05 (11.91)	0.218
Gross motor	41.07 (13.89)	45.00 (10.27)	40.23 (12.29)	0.164
Communication	48.21 (18.97)	54.24 (8.00)	56.14 (5.76)	0.057

* Based on one-way ANOVA.

**Table 6 children-09-01581-t006:** Factors affecting decreased problem-solving domain using regression analysis.

Variable	Problem-Solving *		
	Coeff B	SE (B)	*p*-Value
Newborn vitamin D (ng/mL)	−0.641	0.175	<0.001
Parity	−0.217	2.434	0.929
Maternal occupation	−3.102	3.194	0.334
Maternal education	1.446	2.472	0.560
Gestation age	0.725	0.713	0.312
Exclusive breastfeeding (0 = no; 1 = yes)	−7.099	3.867	0.070
Sex (M = 1; F = 2)	6.345	2.206	0.005
Constant	−13.159	–	

* *R*^2^ = 21.4%.

## Data Availability

Not applicable.

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
