# Peer review of "Correlation between Cord Blood Vitamin D Levels and Problem-Solving Neurodevelopment in Early Childhood: A Cohort Study in Rural Indonesia"

_children, 2022, doi:10.3390/children9101581_

Round 1

Reviewer 1 Report

This is an important study that examines the relationship between vitamin D and neurodevelopmental outcomes. Previous studies have had mixed findings, so additional contributions are helpful in better understanding this relationship. The authors use data from an existing cohort study that has some distinct advantages, with the availability of cord blood samples for vitamin D measurement, and assessment of developmental outcomes at three different infant ages.

Their methods are generally appropriate but require some clarification. In addition, there are some corrections that are simple to make (but important). These are listed below. Not listed are the numerous grammatical errors that should be corrected through a careful proofreading.

- I believe that table 6 (and lines 142-3) reports the result of linear regression (as stated in the statistical methods section) and not logistic regression. (If it is logistic regression, their statistical methods would need to mention where the problem-solving scores were dichotomized, and provide a rationale for doing so.)

- Line 142 mentions the personal-social domain, where I think the authors mean to say problem-solving domain (as that is the one whose relationship is further explored with the regression model).

- Did all 116 infants have complete data, i.e., available vitamin D level, and assessments of all domains at all three time points? If so, this should be stated in the methods. If not, n's should be provided with each analysis (e.g., for each column of tables 3 and 4, and row sections of table 5).

- The conclusion is overly broad, given that their findings are mostly limited to the problem-solving domain, and the statement should be narrowed to reflect what the data actually show. The absence of associations in the other domains is presumably also of interest.

- The description of power (line 114) is unclear and inadequate. The authors should specify what effect size they had power for (as well as the level of power) with the achieved sample size, and for what specific test or analysis (i.e., is it just for the correlation analysis that used continuous data? categorized vitamin D levels had only 15 subjects in the deficient range, and so presumably power is lower for those analyses)

- line 219 is unclear. Limitations are not unreported, as they then go on to name some (they might name others, but it is not the case that they are not reported - perhaps another word was meant there?)

- lines 45-59 are unclear (and have poor grammar). The first sentence needs to be rewritten to be understood. For the second sentence, did they mean to say "As with an association..." (instead of "As if...")? Also, I would replace "it is very likely..." with "it seems likely..." so as not to overstate.

-lines 105-6: SPSS actually stands for Statistical Package for Social Sciences

-line 74: should say "...and of appropriate size for gestational age.." or similar

-line 95: section 2.4 heading should say Statistical Analysis (not Statistic)

Finally, some thoughts on interpretation:

- Are the cutoffs that are used for insufficiency and deficiency, which are used in adults, appropriate for infants? Is it possible that their levels of nutritional need are different? Is there any literature that speaks to this?

- If the effects on neurodevelopment are only evident for those with very low vitamin D levels (i.e., a threshold effect), then the fact that the sample includes primarily subjects across a range of non-deficient levels may dilute the observed effects and reduce power. A study with a greater number of very low levels might be better able to elucidate relationships.

- An experimental design would help, especially given mixed results from observational studies. (The authors allude to this when they suggest that RCTs of supplementation might help.) Are there any animal studies on this question?

Reviewer 2 Report

This manuscript reports a relationship between low vitamin D status of children aged 6 to 24 months and clinical assessment of their neurodevelopment. The effect of low vitamin D status was reported to be significantly linked to poor problem solving skills at this early age. The following points need to be addressed by the authors.

1.      Throughout the text the term vitamin D level has been incorrectly used when referring to the concentration of 25-hydroxyvitamin D in blood serum. This is a confusing use of the term vitamin D level because the concentration of vitamin D, which is also in blood serum as well as 25-hydroxyvitamin D, was not measured. This really needs to be corrected. There are two terms that truly indicate what the authors are measuring. One of these is the concentration of 25-hydroxyvitamin D in blood serum. The other term is vitamin D status. Vitamin D status is determined by the concentration of 25-hydroxyvitamin D in blood serum. It is not determined by measuring the vitamin D levels in blood serum.

2.      In Table 1, the layout seems to have become disrupted as the column Characteristics is not aligned with the numbers in columns labelled Total and %.

3.      Lines 190-192: vitamin D could pass through the placenta and enter the fetal bloodstream with a half-life of approximately 2 months. Firstly, it is not vitamin D that passes from the maternal to the fetal circulation, it is 25-hydroxyvitamin D. Secondly, some reference is needed to support the surprising statement that 25-hydroxyvitamin D in the fetal blood has a half-life of 2 months approximately 60 days. What could explain this long residence time in fetal blood of the 25-hydroxyvitamin D?
